# Regulation of Host Immune Response against *Enterobacter cloacae* Proteins via Computational mRNA Vaccine Design through Transcriptional Modification

**DOI:** 10.3390/microorganisms10081621

**Published:** 2022-08-10

**Authors:** Muhammad Naveed, Khizra Jabeen, Rubina Naz, Muhammad Saad Mughal, Ali A. Rabaan, Muhammed A. Bakhrebah, Fahad M. Alhoshani, Mohammed Aljeldah, Basim R. Al Shammari, Mohammed Alissa, Amal A. Sabour, Rana A. Alaeq, Maha A. Alshiekheid, Mohammed Garout, Mohammed S. Almogbel, Muhammad A. Halwani, Safaa A. Turkistani, Naveed Ahmed

**Affiliations:** 1Department of Biotechnology, Faculty of Sciences and Technology, University of Central Punjab, Lahore 54590, Pakistan; 2Corona Intensive Care Units, District Headquarter Teaching Hospital, Dera Ghazi Khan 33000, Punjab, Pakistan; 3Molecular Diagnostic Laboratory, Johns Hopkins Aramco Healthcare, Dhahran 31311, Saudi Arabia; 4College of Medicine, Alfaisal University, Riyadh 11533, Saudi Arabia; 5Department of Public Health and Nutrition, The University of Haripur, Haripur 22610, Pakistan; 6Life Science and Environment Research Institute, King Abdulaziz City for Science and Technology (KACST), Riyadh 11442, Saudi Arabia; 7Department of Clinical Laboratory Sciences, College of Applied Medical Sciences, University of Hafr Al Batin, Hafr Al Batin 39831, Saudi Arabia; 8Department of Medical Laboratory Sciences, College of Applied Medical Sciences, Prince Sattam bin Abdulaziz University, Al-Kharj 11942, Saudi Arabia; 9Department of Botany and Microbiology, College of Science, King Saud University, Riyadh 11451, Saudi Arabia; 10Department of Medical Laboratories Technology, Faculty of Applied Medical Science, Taibah University, Al Madinah Al Munawarh 344, Saudi Arabia; 11Department of Community Medicine and Health Care for Pilgrims, Faculty of Medicine, Umm Al-Qura University, Makkah 21955, Saudi Arabia; 12Department of Medical Laboratory Sciences, College of Applied Medical Sciences, University of Hail, Hail 4030, Saudi Arabia; 13Department of Medical Microbiology, Faculty of Medicine, Al Baha University, Al Baha 4781, Saudi Arabia; 14Department of Medical Laboratory, Fakeeh College for Medical Science, Jeddah 21134, Saudi Arabia; 15Department of Medical Microbiology and Parasitology, School of Medical Sciences, Universiti Sains Malaysia, Kubang Kerian 16150, Kelantan, Malaysia

**Keywords:** immunoinformatic, in silico, bioinformatics, antibiotic resistance, AMR

## Abstract

*Enterobacter cloacae* is mainly responsible for sepsis, urethritis, and respiratory tract infections. These bacteria may affect the transcription of the host and particularly their immune system by producing changes in their epigenetics. In the present study, four proteins of *Enterobacter cloacae* were used to predict the epitopes for the construction of an mRNA vaccine against *Enterobacter cloacae* infections. In order to generate cellular and humoral responses, various immunoinformatic-based approaches were used for developing the vaccine. The molecular docking analysis was performed for predicting the interaction among the chosen epitopes and corresponding MHC alleles. The vaccine was developed by combining epitopes (thirty-three total), which include the adjuvant Toll-like receptor-4 (TLR4). The constructed vaccine was analyzed and predicted to cover 99.2% of the global population. Additionally, in silico immunological modeling of the vaccination was also carried out. When it enters the cytoplasm of the human (host), the codon is optimized to generate the translated mRNA efficiently. Moreover, the peptide structures were analyzed and docked with TLR-3 and TLR-4. A dynamic simulation predicted the stability of the binding complex. The assumed construct was considered to be a potential candidate for a vaccine against *Enterobacter cloacae* infections. Hence, the proposed construct is suitable for in vitro analyses to validate its effectiveness.

## 1. Introduction

Respiratory tract and urinary tract infections are mostly caused by the pathogen *Enterobacter cloacae*. *Enterobacter* spp. the second most prevalent carbapenem-resistant *Enterobacteriaceae* (CRE) in the United States, has an increasing role in the spread of carbapenem-resistant infections [1]. *Enterobacter cloacae* are anaerobic Gram-negative bacteria involved in neonatal infections that are associated with morbidity and multi-drug resistant infections [2]. It has developed as one of the most prevalent nosocomial bacteria in recent decades, infecting patients with underlying disorders, immunosuppression, and prolonged hospitalization, particularly in the ICU and burns department. The bacterium often causes sepsis, urethritis, and lower respiratory tract infection; however, septic osteoarthritis is unusual [1]. New evidence indicates that it is also a prevalent cause of infections in orthopedic departments [2].

There is intrinsic resistance of *E. cloacae* to amoxicillin, ampicillin, cefoxitin and cephalosporins, regardless of AmpC-lactamase production [3]. There is quite significant prevalence of enzymatic resistance against the broad-spectrum anti-bacterial drug cephalosporins [4,5]. Because AmpC-lactamase overproduction is the most prevalent cause of third-generation cephalosporin resistance, therapy with cephalosporins may promote overproducing AmpC mutants [6]. Fourth-generation cephalosporins have considerable efficacy against de-repressed strains, but if the strains also develop ESBLs, they acquire genes for resistance to this antibiotic class [7].

Microorganisms possess key regulators known as epigenetics, which work as protective mechanisms by altering gene activity without modifying the DNA sequence [8]. Many infectious agents cause epigenetic changes that alter gene transcription in the human host, particularly immune cells. These changes are caused by a variety of byproducts and drugs. These changes comprise RNA-based modifications and DNA methylation. Because they change the genes of host innate and adaptive immune systems, these chemicals are crucial in the treatment of infections [9]. Quaternary ammonium compound efflux SMR transporter SugE (Accession number: A0A1B3EWV7) overexpression increases the resistance to ammonium compounds and functions as an epigenetic performer in modulation for the expression of SugE in *E. cloacae* [10,11]. Treatment with a sulfated derivative of EPS improves intestinal epithelial cell proliferation and persistence, as well as promoter DNA methylation, which controls the expression of mitogen-activated protein kinases (MAPKs) and Janu kinase (JAK2), which alleviate inflammatory damage [2,9].

The primary objective of the current study was to create a new multi-epitope mRNA vaccine employing outer membrane porins (OMPs) of *E. cloacae* that play a role in the epigenome of the host. The vaccine construct was developed using an in silico technique. The cytotoxic T-lymphocyte (CTL), B-cell, and also HT-lymphocyte (HTL) epitopes were obtained by performing computational analysis on the chosen protein sequences. The allergenicity, antigenicity, and toxicity of specific epitopes was accessed. The selected epitopes were also evaluated to check whether they have the ability to develop autoimmunity or not. The docking analysis was performed among epitopes and their corresponding MHC alleles to calculate the vaccine construct population coverage worldwide. Furthermore, this study predicts the validation of the hypothesis by performing an in silico immune simulation. The hierarchical approach for the 3D structure prediction was performed for the peptide sequence of the vaccine design and was docked against TLR-3 and TLR-4. Finally, the stability of the vaccine complex was validated using molecular dynamic modeling.

## 2. Materials and Methods

### 2.1. Sequence Retrieval of Bacterial Protein

The sequence of amino acids of bacterial proteins that are involved in porin activity and ions transmembrane transport were obtained from the database Uniport (http://www.uniprot.org, accessed on 25 November 2021). The bacterial sequences in amino acid form were retrieved through the Uniport database in FASTA format of protein OmpF (Accession Id-V5ISF4), OmpD (Accession Id-A0A2T4Y430), OmpC (Accession Id-Q93K99), and Omp35 (Accession Id-G3LW48). The sequences retrieved were subjected to highly immunogenic spot for the prediction of efficient epitopes [12].

### 2.2. Immunogenic-Antigenic and Allergenic Prediction of Protein

The VaxiJen (http://www.ddg-pharmfac.net/vaxijen/VaxiJen/VaxiJen.html, accessed on 26 November 2021) analysis was performed to predict the immunogenicity or antigenicity of retrieved bacterial protein sequences. The algorithm of Auto Cross Covariance (ACC) was selected to predict the results with an accuracy level up to 89%. The server Aller-TOP (https://www.ddg-pharmfac.net/AllerTOP/index.html, accessed on 26 November 2021) was employed to predict allergenicity of retrieved bacterial protein sequences.

### 2.3. Immunoinformatics Analysis

#### 2.3.1. CLT Epitopes Prediction

The outer membrane protein sequences were analyzed for the screening of CLT epitopes by computational tools of the MHC-I IEDB server (http://tools.iedb.org/mhci/, accessed on 26 November 2021). To execute, the server needed FASTA sequences of chosen proteins. To predict the epitopes, 9 and 10 mer lengths were selected and artificial neural networks (ANN) method was used. For the ultimate prediction results of the epitopes, human complete set HLA was used and sorted by IC50 assessment. The epitopes over 500 IC50 values were selected for further interpretation [13].

#### 2.3.2. Prediction of B-Cells Epitopes

The webserver of ABCpred (https://webs.iiitd.edu.in/raghava/abcpred/ABC_submission.html, accessed on 27 November 2021) was utilized to predict B-cell epitopes. The artificial machine learning approach was performed for epitope prediction. A 0.5 threshold was used to submit each protein sequence. The selected epitope’s length was 16 mer. Moreover, the overlap filter remained active. The top epitope results were chosen for further investigation.

#### 2.3.3. Prediction of HLT-Epitope

An IEDB database MHC-II server (http://tools.iedb.org/mhcii/ accessed on 27 November 2021) was run to obtain information on HTL epitopes in relation to a comprehensive human HLA reference set [14]. The protein sequences were provided as the tool in FASTA format [15]. The epitopes were predicted using the allele-specific approach NN-align 2.3 (Net MHC II 2.3). The length of the epitopes was set at 15 mer [16]. The IFN-epitope (http://crdd.osdd.net/raghava/ifnepitope/predict.php accessed on 27 November 2021), IL-10 pred (https://webs.iiitd.edu.in/raghava/il10pred/predict3.php accessed on 27 November 2021) and IL-4 pred (https://webs.iiitd.edu.in/raghava/il4pred/predict.php accessed on 27 November 2021) online tools were used to predict that the selected epitopes secrete IFN-γ, IL-10 and IL-4, which particularly stimulate interferon–gamma cytokines. This server’s algorithm utilized a hybrid technique that combines the strengths of the motif and SVM models [17].

#### 2.3.4. Human Homology

All predicted peptides were compared to Homo sapiens (TaxID: 9606) protein database performing the BLASTp tool (https://blast.ncbi.nlm.nih.gov/Blast.cgi?PAGE=Proteins accessed on 28 November 2021). If the E-value was more than 0.05, all peptides in the vaccine were considered possible non-homologous peptides.

#### 2.3.5. Antigenicity, Allergenicity and Toxicity Assessment of Epitopes

All particular epitopes were evaluated for antigenicity, allergenicity, and toxicity [18]. To evaluate antigenicity, the VaxiJen antigen server (http://www.ddg-pharmfac.net/Vaxijen/VaxiJen/VaxiJen.html accessed on 1 December 2021) was employed [19]. The prediction in an alignment-independent technique is based on the physicochemical properties of the epitopes. Bacteria having a 0.5 threshold were chosen for analysis. To analyze the allergenicity of epitopes, the website of AllerTop V.2.0 (http://www.ddg-pharmfac.net/AllerTOP accessed on 1 December 2021) was employed. All factors were set to their default values [20]. Finally, the ToxinPred (https://webs.iiitd.edu.in/raghava/toxinpred/multisubmit.php accessed on 1 December 2021) server for toxicity analysis was used to determine and measure the toxicity of the epitopes. Only the antigenic, non-toxic, and non-allergenic epitopes were chosen for additional analysis [21].

#### 2.3.6. Sequences Alignment

The database of NCBI was utilized to obtain all potential variations of specified proteins. The Bio-Edit sequence alignment tool was run for multiple alignment sequence, alignments, and visualization. The conserved areas found epitopes and then further screened them out and used for next interpretation.

#### 2.3.7. T-Lymphocytes and their MHC-Alleles Molecular Docking Analysis 

The molecular docking modeling used for the interpretation of binding affinity of retrieved T-lymphocytes with the MHC alleles. The MHC alleles’ 3D structures were obtained from the PDB-Database. After that, the structures were optimized with the PyMOL tool to remove the unnecessary ligands. Following that, ClusPro (https://cluspro.bu.edu/login.php accessed on 3 December 2021) was utilized for computing binding affinity and to dock each epitope along its associated MHC allele. PyMOL or Discovery Studio were used to assess the orientation and interactions.

#### 2.3.8. Population Coverage by IEDB

The IEDB database’s Coverage of Population webserver (http://tools.iedb.org/population/ accessed on 4 December 2021) was used to compute coverage for vaccine design to target T-lymphocyte epitopes and associated MHCI and MHCII alleles [22]. This calculated value was determined by the coverage of MHC alleles recognized by the construct’s epitopes [23]. This was because the distribution of MHC alleles varies between geographical or ethnic groups throughout the world.

### 2.4. Vaccine Designing 

The following mRNA-based vaccination design was built since the N-terminus to the C-terminus: 5′ m7GCap-5′ UTR-Kozak sequence-tPA (Signal peptide)-EAAAK Linker- Resuscitation-promoting factor (RpfE) (Adjuvant)-GPGPG linker-HTL Epitopes-KK-B-Cell Epitopes-AAY Linker-CTL Epitopes—MITD sequence-Stop codon-3′ UTR-Poly (A) tail.

Three linkers were used to connect all predicted epitopes: GPGPG, KK, and AAY. These linkers exist to divide domains and to allow them to function independently. They can be cleaved, are flexible, and are hard. To increase the adaptive immune response, an adjuvant RpfE was used. A Kozak sequence, which includes a start codon in the ORF and a stop codon, must be included in the mRNA vaccine. In addition, the construct was given two structures: first, the tissue Plasminogen Activator (tPA) secretory signal sequence (UniProt ID: P00750) in the construct’s 5′ region. This is a signal sequence that aids in the secretion of epitopes once they have been translated from cells; second, the MHC I-targeting domain (MITD) (UniProt ID: Q8WV92) in the 3′ locus end of the mRNA vaccine. 

#### 2.4.1. Evaluation of Physiochemical Profiling of Translated Vaccine 

The VaxiJen and ANTIGENpro servers were run for the evaluation of antigenicity of the design. This interpretation was important because antigenicity of an epitope may induce an immunity and memory to generate cell production [24]. VaxiJen 2.0 predicts vaccine qualities based on many physicochemical features, whereas the server of ANTIGENpro (http://scratch.Proteomics.ics.uci.edu/ accessed on 6 December 2021) is established on data obtained through the analysis of microarray and machine learning algorithms. The input of mRNA vaccines consists of only the amino acid sequence, with no tPA or MITD sequences.

The AllerTOP 2.0 server was used for evaluation of the construct’s allergenicity and to ensure that there are not any allergenic factors, and the ToxinPred server was used to assess the vaccine’s toxicity. Finally, to estimate the vaccine’s various physicochemical parameters, ProtParam webserver (https://web.expasy.org/protparam/ accessed on 7 December 2021) was used. Among these characteristics are the amino acid concentration, theoretical isoelectric point (pI), molecular weight, Aliphatic Index (AI), Instability Index (II), and Grand Average of Hydropathicity (GRAVY) [25].

#### 2.4.2. Immune Simulation Response

The server of the C-ImmSim simulation (http://150.146.2.1/C-IMMSIM/index.php accessed on 9 December 2021) was used to evaluate immunological response of the vaccine design while leaving the parameters at default. The mechanism based on the interaction of epitopes with receptors of the lymphocyte impersonated the immunological response. Most contemporary vaccines prescribe two to three doses every four weeks. As a result, in this study’s immunological simulation, three doses of 1000 vaccine units were administered during a four-week period. Three injections were run at distinct time-steps of 1, 84, or 168.

#### 2.4.3. Optimization of Codon of mRNA Vaccine

For effective production within human cells, the peptide vaccine construct must be codon optimized. As a result, GenScript’s (GSs) tool of GenSmart Codon Optimization (http://www.genscript.com/ accessed on 10 December 2021) was used. The optimized sequence’s quality was evaluated using GenScript’s Rare Codon Analysis tools. The Codon Adaptation Index indicates the efficiency of mRNA translation (CAI). Codon Frequency Distribution indicated the presence of any tandem uncommon codons (CFD).

#### 2.4.4. Secondary Structure Prediction for mRNA Vaccine

The RNAfold tool (http://rna.tbi.univie.ac.at/cgi-bin/RNAWebSuite/RNAfold.cgi accessed on 11 December 2021) was used to predict the secondary structure of the mRNA vaccine. It leverages McCaskill’s method to calculate the lowest free energy of the expected secondary structure (MFE). The minimal free energy (MFE) of the structure and the centroid secondary structure, as well as their minimum free energy, were calculated [26].

#### 2.4.5. Estimation and Validation of Peptides Structure

The PSIPRED tool (http://bioinf.cs.ucl.ac.uk/psipred/ accessed on 13 December 2021) was utilized to predict the peptide’s secondary structure using position-specific scoring matrices with an accuracy of 84.2 percent [27]. A peptide sequence’s three-dimensional structure was predicted using the Robetta service (https://robetta.bakerlab.org/ accessed on 14 December 2021). To confirm the optimum structure, ProSA-web (https://prosa.services.came.sbg.ac.at/prosa.php accessed on 14 December 2021), ERRAT (https://saves.mbi.ucla.edu/ accessed on 14 December 2021), and PROCHECK were performed.

#### 2.4.6. Conformation of B-Cells Epitope Prediction

The protein’s tertiary structure can elicit novel conformational B-cell epitopes [28]. ElliPro, an online service (http://tools.iedb.org/ellipro/ accessed on 16 December 2021), was used to estimate the protein structure’s discontinuous B-cell epitopes [29]. Ellipro takes advantage of the 3D model’s geometrical characteristics. When compared to other known strategies for predicting discontinuous B-cell epitopes, ElliPro has the greatest AUC value of 0.732 for the protein model.

#### 2.4.7. Molecular Docking of Vaccine Construct

The vaccine peptide and Toll-like receptor 4 (TLR-4) (PDB ID: 3FXI) or TLR-3 (PDB ID: XI) (PDB ID: 1ZIW) were used for docking. The PIPER docking technique was used to dock 3D buildings on the ClusPro server. A RpfE adjuvant was docked against TLR4 and TLR3 as a control. This server may generate many models based on different scoring methods. The PRODIGYY tool on the HADDOCK website (https://haddock.science.uu.nl/ accessed on 17 December 2021) was used to calculate the free dissociation constant (Kd), binding energy (G), and percentages (%) of polar and charged amino acids on the non-interacting surface by receptor–ligand 3D interaction [30]. 

#### 2.4.8. Molecular Dynamic Simulation

The iMODS server (http://imods.chaconlab.org/ accessed on 20 December 2021) executed dynamics simulation analysis for the TLR4-vaccine and TLR3-vaccine complex structures with the lowest binding energy to verify the mobility and stability of the vaccine’s complex structures and molecules.

## 3. Results

### 3.1. B-Cell Epitopes Prediction and Evaluation

As shown in Table 1, the targeted proteins from *Enterobacter cloacae* bacteria were chosen based on their antigenicity and allergenicity, as well as their molecular and biological functions.

From each targeted protein sequence, we selected the top three epitopes by using the ABCpred server. Further, we screened only epitopes that have antigenic, non-toxic, non-homologous, and non-allergenic properties for vaccine construction. Moreover, all screened epitopes were analyzed if they were homologous to Homo sapiens, to eliminate from the vaccine design those that may introduce autoimmunity. Table 2 shows that we chose eight B-cell epitopes commencing the targeted proteins to include in the vaccine design.

### 3.2. CTL Epitopes Prediction and Evaluation

From the targeted four proteins, we chose epitopes of CTL from the IEDB’s database of MHC-I. The parameter used included IC50 over 500 and the ANN predicted method to select the epitopes. Furthermore, we exclusively chose antigenic, allergenic, non-toxic and non-homologous epitopes. As a consequence, as shown in Table 2, 18 epitopes were chosen for vaccine construction.

### 3.3. HTL Epitopes Prediction and Evaluation

*Enterobacter cloacae* outer membrane proteins were evaluated to identify potential HTL epitopes. Only epitopes investigated as antigenic, non-toxic, non-allergenic and lastly non-homologous and that have the ability to induce INF-gamma, IL-4, and IL-10 were selected. As shown in Table 2, eight epitopes that may trigger cytokines and that exit in a conserved area were chosen.

### 3.4. Molecular Docking Interaction of Epitopes and MHC-Alleles

There was a total epitope count of 26 T-lymphocyte with 92 MHC alleles in correspondence. Some epitopes recognize only one MHC allele, whereas others choose up to ten MHC alleles, as shown in Table 3. We chose five epitopes with corresponding MHC alleles from among them for molecular docking investigation. Table 4 presents the docking outcomes from the server of ClusPro 2.0 in the form of energy affinity. The QNGNKTRLAFAGLKF epitopes with their corresponding MHC alleles (HLA-DRB*15:01) show the strongest binding affinity at −698.6 kcal/mol. Consequently, epitopes bind to MHC allele binding forks competently, as shown in Figure 1A,B. Furthermore, all the possible interactions of targeted epitopes and the residues of alleles were evaluated as shown in Figure 2. 

### 3.5. Vaccine Construct

The proposed construct of the mRNA-based vaccine illustrated in Figure 3 displays the vaccine from the N-terminal to C-terminal, which is: 5′ m7GCap-5′ UTR-Kozak sequence-tPA (Signal peptide)-EAAAK Linker-RpfE (Adjuvant)-GPGPG linker-QNGNKTRLAFAGLKF-GPGPG-VAQYQFDFGLRPSIA-GPGPG-YFNKNMSTYVDYKIN-GPGPG-NKNMSTYVDYKINLL-GPGPG-NIYLASTYSETRNMT-GPGPG-QNGNKTRLAFAGLKF-GPGPG-NGNKTRLAFAGLKFG-GPGPG-YFNKNMSTYVDYKIN-KK-GLHYFSDNDSNDGDNT-KK-AGAANAAEIYNKDGNK-KK-YIDVGATYYFNKNMST-KK-IGDEDYINYIDVGATY-KK-SGYGQWEYEFKGNNDE-KK-AGVVNAAEIYNKDGNK-KK-PEFGGDTYGSDNFMQQ-KK-YGQWEYQIQGNSGENE-AAY-DNTYARLGFK-AAY-YGKAVGLHYF-AAY-AITSSLAVPV-AAY-NTYARLGFK-AAY-NTYARLGFK-AAY-TGYGQWEYNF-AAY-WATSLSYDF-AAY-AQYQFDFGL-AAY-MSTYVDYQIN-AAY-KTYVRLGFK-AAY-MSTYVDYKI-AAY-SGYGQWEYEF-AAY-AQYQGKNNK-AAY-GYGQWEYEF-AAY-MSTYVDYKI-AAY-KVLSLLVPAL-AAY-KYVDVGATYY-AAY-FGLRPSVAYL-AAY-VLSLLVPALHTL-MITD Sequence-Stop codon-3′ UTR-Poly (A) tail.

### 3.6. Assessment of Physiochemical Profiling of Vaccine Design

The ExPasy ProtParam service was used to determine physiochemical profile of the construct as illustrated in Table 5. It was expected that the vaccine would be non-allergenic, antigenic, non-toxic, and soluble. All the physiochemical properties of the vaccine construct predicted that the construct was thermally stable. The GRAVY measurement, which was −0.456, shows a hydrophilic nature of the vaccine construct. Thus, profiling of a multi-epitope mRNA-based vaccine reveals that it has the potential to be a vaccination candidate.

### 3.7. Population Coverage Prediction

The coverage of the population of the world was predicted by combining MHC-I and II and the total amount of 92 alleles of the corresponding 26 epitopes with the tool of IEDB population coverage. Finally, vaccine coverage was predicted to be around 99.2% worldwide. 

### 3.8. Immune Simulation Response 

Three vaccine injections were given to boost the response of immunity against pathogenic bacterium, as second and third injections have higher responses than the first one. Appendix A illustrates that immunoglobulin (IgM) production is higher than IgG and the antigenic response, in terms of the level of immunoglobulin increase. This highlights the immune memory emergence from antigen exposure. There was a boost in CTL and HTL via the generation of memory cells. Moreover, microphage activity was boosted, whereas dendritic cell activity was constant. Finally, the Simpson index is low, which shows the production of interleukins and cytokines.

### 3.9. Codon Optimization of Construct

The GenSmart Codon Optimization tool was used to improve the translation of an mRNA vaccination within the host. The length of the mRNA vaccine consists of 2151 nucleotides. The GC content’s optimal percentage should be in the choice of 30–70 for efficient expression of the vaccine inside the human cell. The average GC content percentage of this construct was 58.44%.

### 3.10. mRNA Vaccine’s Secondary Structure Prediction

The webserver of the RNA-fold was used to examine the vaccine’s mRNA structure. Free energy assessment was performed by this server. For input, we used an adjusted codon of the vaccine design. Finally, the mRNA will stabilize, which generates a structure as shown in Figure 4, with a minimum free energy of −646.75 kcal/mop and a secondary centroid structure of −545.77 kcal/mol. The mRNA secondary structure was predicted to improve the stability of mRNA against endonuclease cleavage and chemical degradation.

### 3.11. Peptides Structure Prediction of mRNA Vaccine

The online server PSIPRED’s was exploited to construct the vaccine’s secondary structure as presented in Appendix A. Furthermore, Appendix A shows that the Robetta server predicted the tertiary structure. The overall quality factor was 89.172. The Ramachandran plot was constructed to indicate that 96.8% of the mostly favored region is displayed in Appendix A. The ProSA-web predicted a negative Z-score, presented in Appendix A, which is about −5.46 and which shows consistency of the structure.

### 3.12. Conformational Prediction of B-Cell

The server ElliPro was used for folding the vaccine model to identify B-cell conformational epitopes. This tool was used to predict eleven discontinuous conformational B-cell epitopes. Appendix A shows the secondary or tertiary models of conformational cell epitopes, which have a total of 365 residues with the predicted score ranging from 0.507 to 0.816.

### 3.13. Molecular Docking of Vaccine Peptides

The ClusPro server was used to perform docking studies between the vaccination and the TLR-3 and TLR-4 immunological receptors as displayed in Figure 5A and Figure 6A. In addition, the PRODIGY server was utilized to anticipate the binding affinities along with dissociation constants at 37 degrees Celsius for each complex independently. TLR-3 and TLR-4 immune receptors docked the adjuvant in the control group. For the TLR3–vaccine complex, −16.6 kcal/mol^−1^ was the binding affinity, while for the control, it had a −8.1 kcal/mol^−1^ binding affinity. The complex dissociation constant at 25C for vaccine–TLR3 was 7.2 × 10^13^ as compared to the control, which had a 1.1 × 10^−6^ value. In the case of the TLR4–vaccine complex, −18.7 kcal/mol^−1^ was the binding affinity, while for the control, it had a −9.0 kcal/mol^−1^ binding affinity. The complex dissociation constant at 25C for vaccine–TLR4 was 2.1 × 10^14^ as compared to the control with a 2.5 × 10^7^ value.

### 3.14. Molecular Dynamic Simulation

The iMOD server was used for the molecular dynamic simulation analysis, when the TLR3 and TLR4 complexes were exposed to the server. The graph of deformability peaks represents the construct’s deformable loci as shown in Figure 5E and Figure 6E, where it presents the coil-shaped amino acids. The B-factor graph depicts the complex’s link between the Normal Mode Analysis and PDB regions as shown in Figure 5F. Figure 5D and Figure 6F show the Eigenvalues of both docked complexes. A covariance matrix depicts the relationships among amino acid duplets in the dynamical area as shown in Figure 5B and Figure 6B, where the red part indicates the correlated residues, the white part shows the anti-correlated residues, and the blue part represents the non-correlated residues.

In Figure 5C and Figure 6C, the model of an elastic network is represented to order the pair of atoms linked with the springs. The grey color represents the stiffer region. Thus, the results indicate that the vaccine construct was found be stiffer and more stable.

## 4. Discussion

*Enterobacter cloacae* has become one of the most emerging and drug-resistant bacteria. There are many anti-bacterial treatments for infections caused by *Enterobacter*, but their resistance makes it a more virulent and emerging bacterium [3]. Despite this, there is no vaccine available for this bacterial infection, and there is need to develop an efficient and effective vaccine to combat against these bacterial infections [1]. The current study was conducted to use immunoinformatics approaches to develop a safe, engineered and efficient vaccine. The mRNA-based vaccine was found to be effective against many viral infections such as Zika virus, influenza virus, rabies, coronavirus and many others [23]. One disadvantage of their use is their instability because of RNase degradation, which is common, and another disadvantage is their innate immunity, which immediately recognizes these structures as foreign [24].

*Enterobacter* and its epigenetics changes have been found link during infection in many in vitro studies [3,6]. The epigenome regulator functions are used to thrive inside the host organisms [31,32]. Subsequently, novel techniques that target function have been created to produce a vaccine or treatment against the infection. These epigenetic changes have an impact on immune system genes as well as on “trained immunity.” According to this notion, an increase in innate immune cells results in fast and strong immunity, which occurs upon second exposure to bacterial infection [33].

Producing long-lasting memory was the main purpose in developing a vaccine. To realize this production, both B and T cells should be activated at same time, which is critical. Consequently, if infectious agents are encountered again in future, the host will respond effectively and quickly. However, the appropriate use of certain antigens known as epitopes is critical for the efficacy of the vaccine. Thus, it is essential to identify the epitope that may stimulate both cells to merge into a vaccine design. The HTL can generate IL-4, IFN-γ and IL-10. Antigen-presenting cells (APCs) shows epitopes of HTL, and the lymphocyte can secrete the chemokines that perform critical functions against these bacteria. Excluding memory cells, all immune cells die after infection is eradicated. The membrane-bound immunoglobin receptor of B-cells is used to recognize the antigenic epitopes. Consequently, it induces the epitopes and processes them before presenting them against T-cells through MHC class-II molecules. These are recognized by T-cell receptors (TCR). As a result, plasma cells differentiate from B-cells and secrete antibodies by neutralizing memory and intruder cells.

To combat infection crises, we developed an in silico based multi-epitope mRNA vaccine based on *Enterobacter cloacae* proteins. These proteins affect the outer membrane of the host organisms. The target proteins are examined for epitopes, which might induce responses of humoral or cell mediation. The web-based IEDB database that predicts epitopes of HTL and CTL are based on immune epitope determination. ABCpred, an internet service, was used to anticipate B-cell epitopes. This server used an artificial machine learning method to predict the epitopes. The evaluation of epitopes was assessed by determining antigenicity, allergenicity and toxicity by using web servers. Specific linkers were used to combine the epitopes together. To boost immunity, the TLR4 immune receptor was added at the N-terminus of the construct. Moreover, an immune simulation of vaccine effects was carried out to validate the humoral and cellular responses of the vaccine. 

The vaccine construct’s targeted epitopes have 92 corresponding MHC alleles. The five chosen epitopes were further exposed for molecular docking. Since the interaction of ligands and epitopes is essential in vaccine design, docking analysis was performed among the chosen epitopes and their corresponding MHC alleles. The ClusPro predicts binding affinity and bond formation of the chosen epitopes. When the energy values are lowest, the receptor strongly binds to the ligands. The interactions among the epitopes and MHC pockets were also analyzed. Furthermore, because there are over one thousand different human MHC alleles, vaccination is only effective in those who have a certain MHC allele that binds the epitope. As a result, the tool of IEDB population coverage predicted that the vaccination would cover 99.2% of the world’s population.

Several types of structures were used to improve stability and translation in the mRNA-based construct. These are the 5′ m7G cap sequences, Globin 5′ and 3′ UTRs bordering the mRNA ORF, Poly (A) tails varying in length from 120 to 150 bps, the Kozak sequence, and the stop codon. In addition, to boost the construct’s efficacy and allocation, the MITD sequence and the tPA secretory signal sequence were inserted to drive it into the endoplasmic reticulum [9,34]. 

Furthermore, the TLR-3 and TLR-4 immune receptors had been docked with the vaccine construct to evaluate the vaccination’s capacity to bind to or to interact by immune receptors. According to the findings, the vaccine has a strong affinity for binding to TLR-4 and TLR-3. As a result, there is a possibility of developing innate and adaptive immunity. The complex’s stability was investigated using molecular dynamic simulation. The mRNA vaccine may be utilized when deprived of an adjuvant. This method has both advantages and disadvantages. By revealing the sequence to annihilation in a wet-lab study, it may save time and money. As a result, an applied study is required to determine which step is necessary.

The recommended mRNA vaccine’s peptide sequence was demonstrated to be stable, thermostable, antigenic, non-allergenic, as well hydrophilic using immunoinformatics approaches. Once delivered in silico with three injections, it was able to elicit an immunological response [35]. It was shown that it could generate a memory cell after being exposed and can produce chemokines that promote B-cell response and humoral response. The generation of memory cells was indicated by macrophages, dendritic cells and by the Simson Index. Finally, the vaccine design is a possible candidate against *Enterobacter* infections.

## 5. Conclusions

The vaccine construct shows appropriate physiochemical and immunological responses. The immunological simulation demonstrated that the vaccination induced an immunity response that was reliable with our objectives. Hence, it is advised that this construct is used as a prospective applicant for in vitro and in vivo studies in contraindications to *Enterobacter cloacae*, with multiple serological analyses used to validate the trigger of response in demand.

## Figures and Tables

**Figure 1 microorganisms-10-01621-f001:**
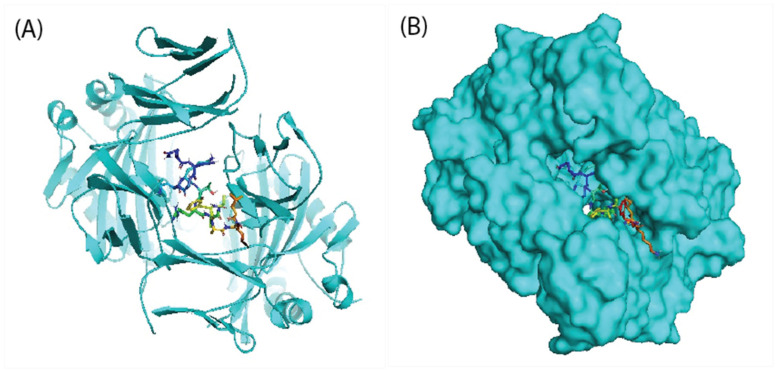
Docking visualization between the epitope QNGNKTRLAFAGLFK and its corresponding MHC allele (HLA-DRB1*15:01) using the software PyMol: (**A**) cartoon view; (**B**) surface view.

**Figure 2 microorganisms-10-01621-f002:**
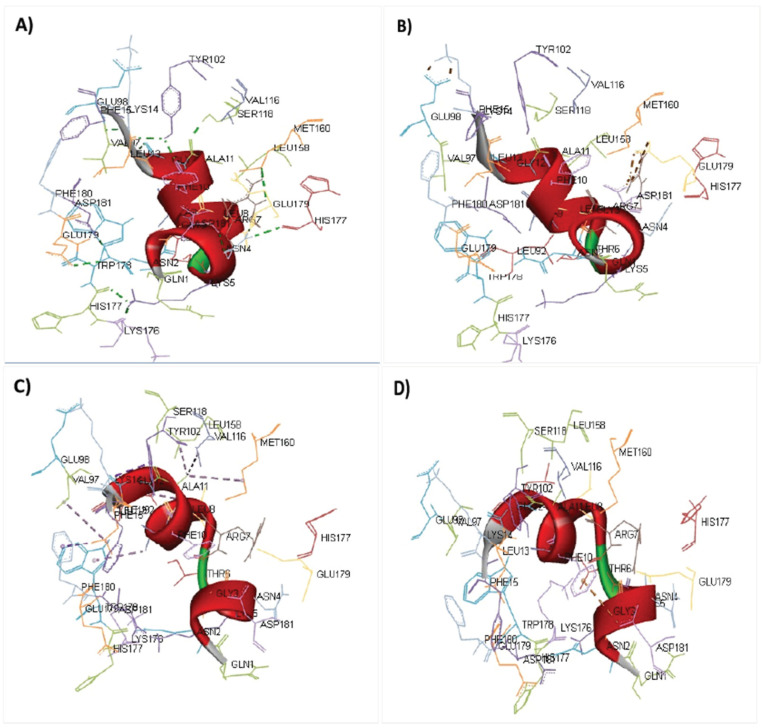
Discovery studio visualization of the different interactions between the epitope and its associated MHC allele: (**A**) conventional hydrogen bonds; (**B**) salt bridge, attractive charge interactions; (**C**) hydrophobic interactions; (**D**) cation–Pi interactions.

**Figure 3 microorganisms-10-01621-f003:**
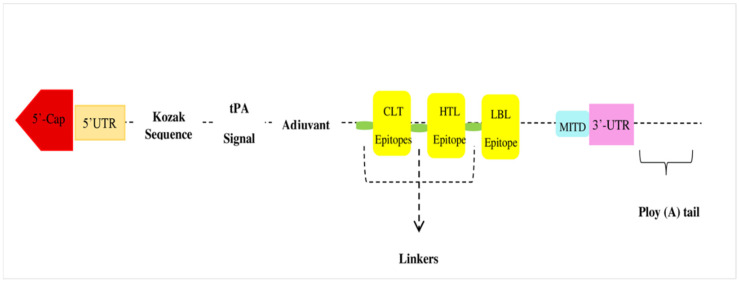
Flow diagram of vaccine construct from N-terminal to C-terminal.

**Figure 4 microorganisms-10-01621-f004:**
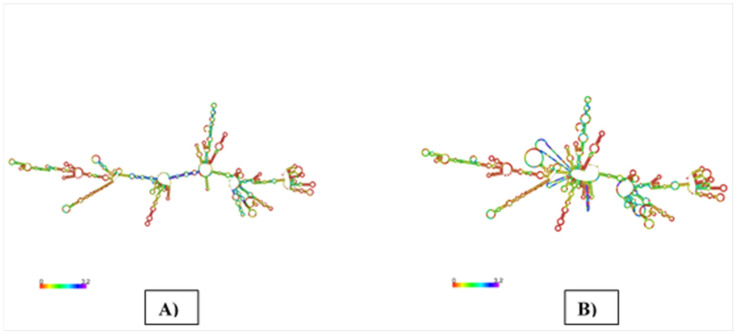
mRNA structure prediction: (**A**) optimal secondary structure; (**B**) centroid secondary structure of the vaccine mRNA retrieved using an RNA-fold internet-based server.

**Figure 5 microorganisms-10-01621-f005:**
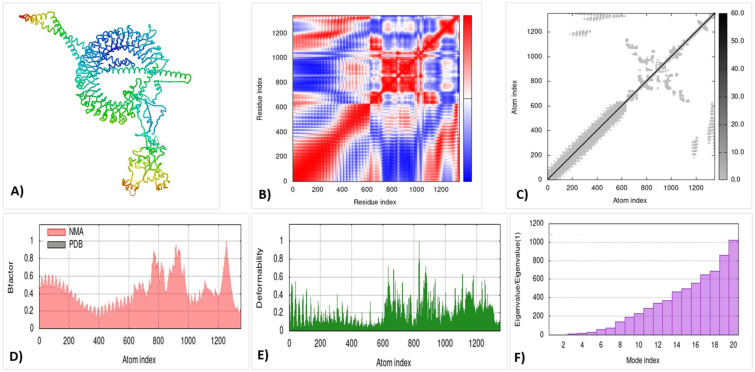
The receptor–ligand interactions, normal mode analysis, and molecular dynamics simulation: (**A**) vaccine–TLR3 docked complex using the ClusPro server; (**B**) covariance matrix; (**C**) elastic network model using the iMODS server; (**D**) B-factor graph; (**E**) deformability graph; (**F**) Eigenvalue of the vaccine–TLR4 complex.

**Figure 6 microorganisms-10-01621-f006:**
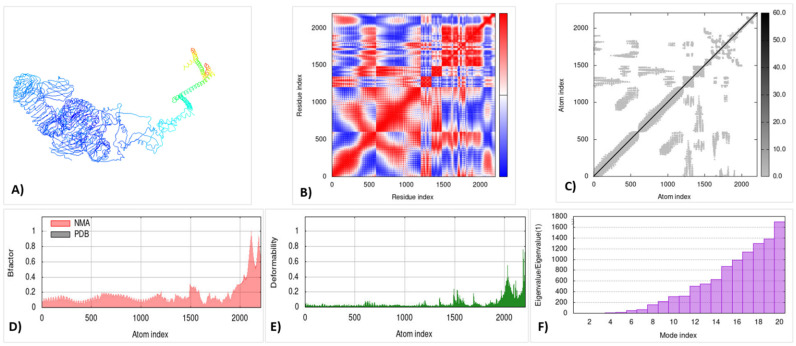
The receptor–ligand interactions, normal mode analysis, and molecular dynamics simulation: (**A**) vaccine–TLR-4 docked complex using the Cluspro server; (**B**) covariance matrix; (**C**) elastic network model using the iMODS server; (**D**) B-factor graph; (**E**) deformability graph; (**F**) Eigenvalue of the vaccine–TLR4 complex.

**Table 1 microorganisms-10-01621-t001:** Targeted proteins and their antigenicity, allergenicity and functions.

Protein *	UniPort Id **	A. Score ***
OmpF	V5ISF4	0.6395
OmpD	A0A2T4Y430	0.6660
Omp35	G3LW48	0.6882
OmpC	Q93K99	0.7882

* Molecular function: porin activity. ** Biological function: ion transmembrane transport. *** Allergenicity: non-allergic.

**Table 2 microorganisms-10-01621-t002:** List of epitope candidates for vaccine design.

Cell Type	Sequence of Epitope
HTL	QNGNKTRLAFAGLKF
	VAQYQFDFGLRPSIA
	YFNKNMSTYVDYKIN
	NKNMSTYVDYKINLL
	NIYLASTYSETRNMT
	QNGNKTRLAFAGLKF
	NGNKTRLAFAGLKFG
	YFNKNMSTYVDYKIN
CTL	DNTYARLGFK
	YGKAVGLHYF
	AITSSLAVPV
	NTYARLGFK
	TGYGQWEYNF
	WATSLSYDF
	AQYQFDFGL
	MSTYVDYQIN
	KTYVRLGFK
	MSTYVDYKI
	SGYGQWEYEF
	AQYQGKNNK
	GYGQWEYEF
	MSTYVDYKI
	KVLSLLVPAL
	KYVDVGATYY
	FGLRPSVAYL
	VLSLLVPAL
B Lymphocytes	GLHYFSDNDSNDGDNT
	AGAANAAEIYNKDGNK
	YIDVGATYYFNKNMST
	IGDEDYINYIDVGATY
	SGYGQWEYEFKGNNDE
	AGVVNAAEIYNKDGNK
	PEFGGDTYGSDNFMQQ
	YGQWEYQIQGNSGENE

**Table 3 microorganisms-10-01621-t003:** T-lymphocytes and their associated alleles.

Protein	CLT Epitopes	MHC-I Binding Alleles	HLT Epitope	MHC-II Binding Alleles
OmpF	AQYQFDFGL	HLA-A*02:06	VAQYQFDFGLRPSIA	HLA-DRB3*01:01, HLA-DRB1*04:05, HLA-DRB1*01:01, HLA-DRB1*03:01, HLA-DRB1*04:01
	KSKAKDVEG	HLA-A*30:01	QNGNKTRLAFAGLKF	HLA-DRB5*01:01, HLA-DQA1*01:02/DQB1*06:02
	WATSLSYDF	HLA-B*35:01, HLA-B*53:01		
	MSTYVDYQIN	HLA-B*58:01, HLA-A*68:02		
OmpD	AQYQGKNNK	HLA-A*11:01	YFNKNMSTYVDYKIN	HLA-DRB1*15:01, HLA-DRB1*07:01, HLA-DRB1*13:02, HLA-DRB1*09:01, HLA-DRB3*01:01, HLA-DRB1*01:01, HLA-DRB1*04:05
	GYGQWEYEF	HLA-A*23:01	NIYLASTYSETRNMT	HLA-DRB1*04:05 HLA-DRB1*08:02 HLA-DRB1*04:01 HLA-DRB3*02:02
	KTYVRLGFK	HLA-A*30:01, HLA-A*03:01, HLA-A*11:01, HLA-A*31:01, HLA-A*68:01		
	MSTYVDYKI	HLA-A*68:02, HLA-B*58:01, HLA-B*53:01		
	SGYGQWEYEF	HLA-A*23:01, HLA-A*24:02		
Omp35	DNTYARLGFK	HLA-A*11:01, HLA-A*03:01	QNGNKTRLAFAGLKF	HLA-DRB5*01:01, HLA-DQA1*01:02/DQB1*06:02, HLA-DRB1*09:01, HLA-DRB1*15:01, HLA-DRB1*07:01, HLA-DRB1*01:01
	YGKAVGLHYF	HLA-B*15:01, HLA-A*23:01	NGNKTRLAFAGLKFG	HLA-DPA1*01:03/DPB1*02:01, HLA-DRB5*01:01, HLA-DPA1*01:03/DPB1*02:01, HLA-DRB1*15:01, HLA-DRB1*09:01, HLA-DRB1*07:01, HLA-DRB1*11:01, HLA-DRB1*01:01
	AITSSLAVPV	HLA-A*02:03, HLA-A*02:06, HLA-A*68:02, HLA-A*02:01		
	NTYARLGFK	HLA-A*68:01, HLA-A*03:01, HLA-A*11:01, HLA-A*30:01, HLA-A*33:01, HLA-A*31:01, HLA-A*26:01		
OmpC	KVLSLLVPAL	HLA-A*02:01, HLA-A*02:06	YFNKNMSTYVDYKIN	HLA-DRB1*15:01, HLA-DQA1*01:01/DQB1*05:01, HLA-DRB3*02:02, HLA-DRB1*07:01, HLA-DRB1*13:02, HLA-DRB1*09:01, HLA-DRB3*01:01, HLA-DRB1*01:01, HLA-DRB1*04:05
	MSTYVDYKI	HLA-A*68:02, HLA-B*58:01, HLA-B*53:01	NKNMSTYVDYKINLL	HLA-DRB1*15:0, HLA-DRB1*03:01, HLA-DPA1*03:01/DPB1*04:02, HLA-DPA1*01:03/DPB1*02:01
	KYVDVGATYY	HLA-A*01:01, HLA-A*30:02		
	FGLRPSVAYL	HLA-A*02:03, HLA-A*02:01, HLA-B*15:01, HLA-A*02:06		
	VLSLLVPAL	HLA-A*02:01, HLA-A*02:03, HLA-A*02:06		

* Proposed peptides.

**Table 4 microorganisms-10-01621-t004:** Molecular Docking of T-lymphocyte epitopes with corresponding MHC alleles and their binding affinities.

Type of Lymphocytes	Epitopes	Alleles	PDB Id	Binding Affinity (kcal/mol)
HTL	YFNKNMSTYVDYKIN	HLA-DRB1*01:01	2FSE	−666.9
	QNGNKTRLAFAGLKF	HLA-DRB1*15:01	1BX2	−698.6
CLT	DNTYARLGFK	HLA-A*11:01	6ID4	−562.0
	KVLSLLVPAL	HLA-A*02:06	3OXR	−544.5
	WATSLSYDF	HLA-B*35:01	4PR5	−569.8

* Proposed peptides.

**Table 5 microorganisms-10-01621-t005:** Physiochemical profiling of the mRNA vaccine.

Physiochemical Profiling	Measurement	Indication
Number of Amino Acid	717	Appropriate
Number of Atoms	10,708	-
Molecular Weight	77,612.39	Appropriate
Formula	C3536H5202N910O1048S12	-
Theoretical pI	8.75	Basic
Total number of negatively charged residues (Asp + Glu)	59	-
Total number of positively charged residues (Arg + Lys)	67	-
Instability Index (II)	29.53	Stable
Aliphatic index	60.08	Thermostable
Grand average of hydropathicity (GRAVY)	−0.456	Hydrophilic
Antigenicity (by VaxiJen)	0.8371	Antigenic
Antigenicity (by ANTIGENpro)	0.802068	Antigenic
Allergenicity	Non-Allergenic	Non-Allergen
Toxicity	Non-Toxic	Non-Toxic
Solubility (m/L)	0.591434	Soluble

## Data Availability

Data will be available upon a resonable request to dr.naveed@ucp.edu.pk.

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
