# Peer review of "Regulation of Host Immune Response against Enterobacter cloacae Proteins via Computational mRNA Vaccine Design through Transcriptional Modification"

_microorganisms, 2022, doi:10.3390/microorganisms10081621_

Round 1

Reviewer 1 Report

The present manuscript describes “Regulation of host immune response against enterobacter cloacae proteins via computational mRNA vaccine design through transcriptional modification” Respiratory tract and urinary tract are infections caused by the pathogen Enterobacter cloacae. The Enterobacter cloacae till now become one of the most emerging bacteria. There were many anti-bacterial treatments for infections caused by Enterobacter cloacae but their resistance made it more virulent and emerging bacteria. Despite this there is no vaccine available for this bacterial infection, there is need to develop the efficient and effective vaccine to combat against the bacterial infections. In order to develop mRNA vaccine against the Enterobacter cloacae infections, the authors have used four proteins of Enterobacter cloacae to predict the CLT, B-cells, HLT, Antigenicity, Allergenicity, and Toxicity assessment of epitopes. The authors evaluated physiochemical profiling of translated vaccine, immune simulation response, and optimized codon of mRNA vaccine. The vaccine construct showed appropriate physiochemical and immunological response. The immunological simulation demonstrated that the vaccination induced an immunity response that was reliable with our objectives. Thus, it is advised that this construct be used as a prospective applicant for in vitro and in vivo research in contradiction of Enterobacter cloacae. I would recommend to publish this work in Microorganisms Journal.

Author Response

Reviewer 1

Comments and Suggestions for Authors

The present manuscript describes “Regulation of host immune response against enterobacter cloacae proteins via computational mRNA vaccine design through transcriptional modification” Respiratory tract and urinary tract are infections caused by the pathogen Enterobacter cloacae. The Enterobacter cloacae till now become one of the most emerging bacteria. There were many anti-bacterial treatments for infections caused by Enterobacter cloacae but their resistance made it more virulent and emerging bacteria. Despite this there is no vaccine available for this bacterial infection, there is need to develop the efficient and effective vaccine to combat against the bacterial infections. In order to develop mRNA vaccine against the Enterobacter cloacae infections, the authors have used four proteins of Enterobacter cloacae to predict the CLT, B-cells, HLT, Antigenicity, Allergenicity, and Toxicity assessment of epitopes. The authors evaluated physiochemical profiling of translated vaccine, immune simulation response, and optimized codon of mRNA vaccine. The vaccine construct showed appropriate physiochemical and immunological response. The immunological simulation demonstrated that the vaccination induced an immunity response that was reliable with our objectives. Thus, it is advised that this construct be used as a prospective applicant for in vitro and in vivo research in contradiction of Enterobacter cloacae. I would recommend to publish this work in Microorganisms Journal.

Response: Dear Reviewer, we would like to really appreciate your kind comments on our manuscript. We hope that we can work together to fight AMR in the future. Again, thank you for your time to review our manuscript and recommend it for possible publication in the journal microorganism.

Reviewer 2 Report

Regulation of Host Immune Response Against Enterobacter Cloacae Proteins via Computational mRNA Vaccine Design 3 Through Transcriptional Modification 

By Naveed et al.

In this paper four proteins of Enterobacter cloacae are used to predict the epitopes for the construction of mRNA vaccine against the bacteria.  This is an important topic considering that bacteria have developed resistance to most antibiotics. However, the paper is written in very poor English language. Initially I tried to correct some of the language but then gave up doing that because there are numerous errors throughout the manuscript.  I strongly suggest that the authors that the help of an expert in English language or pay an appropriate agency to make the English presentable.  Here are some of the language corrections I made initially plus a few more comments.

Line 59: Change “are infections” to “infections are”

Line 59: Change “caused by” to “caused mostly by”

Line 60: Change “spp;” to “spp,”

Line 63: Change “infection” to “infections”

Line 68: Change “However, new” to “New”

Line 68: Change “infection” to “infections”

Line 70-71: This sentence is lacking a verb. Possible correction: Change “The intrinsic ” to “There is intrinsic”

Line 72: Change “with broad-spectrum” to “against the broad-spectrum”

Line 72: Change “product” to “drug” or “compound” or “antibiotic”

Line 73-75: This sentence is lacking a verb. Possible correction: Change “which may excellent for” to “may promote”

Line 76: Change “develop” to “produce” or “acquire genes for”

Line 83: Change “genes of host of their innate” to “genes of the host innate”

Line 84: “Dcm methylated cytosine (accession number: A0A855M2N2) possess DNA-methyltransferase activity.” One cytosine molecule cannot possess DNA-methyltransferase activity. The authors need to mention which gene has this Dcm methylated cytosine.

Line 86: Delete comma (,) before overexpression 

Line 87: Change “resistance of” to “resistance to” or “resistance against”

Line 86-88: This sentence is not clear. Suggestion: Change “have significant part of epigenetic performers” to “functions as an epigenetic performer”

Line 91: Change “alleviates”. If this refers only to JAK2, then this is correct. If it refers to both MAPKs and JAK2, then it should be “alleviate”

Line 93: Change “In this study, it is depicted that clearly that the focus of this article aims” to “This article aims”

Line 95: Change “play role” to “play a role”

Line 95: Change “Purely, the vaccine” to “The vaccine”

Line 96-98: “By doing computational analysis on the chosen protein sequences, the cytotoxic T-lymphocytes (CTL), B-cells, also HT-lymphocytes (HTL) epitopes are obtained.” This sentence is not clear. Suggestion: Change to “The cytotoxic T-lymphocyte (CTL), B-cell, and also HT-lymphocyte (HTL) epitopes were obtained by doing computational analysis on the chosen protein sequences.”

Line 101-102: “Furthermore, … immunogenicity.” It is not clear what the verb is in this sentence. Is “agonists” used as a verb?

Line 102-104: “The docking….worldwide” This sentence does not resemble English language.

Line 105: Change “evaluating” to “evaluated”

I am stopping correcting English language from this point. There are errors in every line. This is affecting the quality of the paper because it is difficult to understand what the authors are trying to say. Numerous sentences don’t have any verb.

Line 280: If the same footnote (*, ** or (***) applies to all cells in a column it can be applied to the column heading only.

Line 348: Table 5: While all other numbers in the table do not have any unit, for solubility, the unit needs to be specified.

Line 372-373: Authors please comment on the significance of these secondary structures and their implications regarding usefulness as vaccines. 

Line 435: “many viral infections” Why is coronavirus, the causative agent of the current pandemic, not in the list? Also, is Ref 10 the right reference for this? It is not clear from the title of the reference.

Author Response

Reviewer 2

Comments and Suggestions for Authors

Regulation of Host Immune Response Against Enterobacter Cloacae Proteins via Computational mRNA Vaccine Design 3 Through Transcriptional Modification By Naveed et al.

In this paper four proteins of Enterobacter cloacae are used to predict the epitopes for the construction of mRNA vaccine against the bacteria.  This is an important topic considering that bacteria have developed resistance to most antibiotics. However, the paper is written in very poor English language. Initially I tried to correct some of the language but then gave up doing that because there are numerous errors throughout the manuscript.  I strongly suggest that the authors that the help of an expert in English language or pay an appropriate agency to make the English presentable.  Here are some of the language corrections I made initially plus a few more comments.

Response: Dear Reviewer, we would like to appreciate your efforts for our manuscript, which make it more suitable for the respective journal, scientific community and readers. We have addressed all of your comments. Furthermore, the manuscript has been thoroughly revised for English proofreading. We also appreciate the way you give us your comments, line by line with recommended suggestion/corrections.

Line 59: Change “are infections” to “infections are”

Response: Line 59: Corrected as “infections are”.

Line 59: Change “caused by” to “caused mostly by”

Response: Line 59: Corrected as “caused mostly by”.

Line 60: Change “spp;” to “spp,”

Response: Line 60: Corrected as “spp,”.

Line 63: Change “infection” to “infections”

Response: Line 63: Corrected as “infections”.

Line 68: Change “However, new” to “New”

Response: Line 68: Corrected as “New”.

Line 68: Change “infection” to “infections”

Response: Line 68-69: Corrected as “infections”.

Line 70-71: This sentence is lacking a verb. Possible correction: Change “The intrinsic ” to “There is intrinsic”

Response: Line 70: Corrected as “There is intrinsic”.

Line 72: Change “with broad-spectrum” to “against the broad-spectrum”

Response: Line 72: Corrected as “against the broad-spectrum”

Line 72: Change “product” to “drug” or “compound” or “antibiotic”

Response: Line 72: Corrected as “drug”.

Line 73-75: This sentence is lacking a verb. Possible correction: Change “which may excellent for” to “may promote”

Response: Line 74: Corrected as “may promote”

Line 76: Change “develop” to “produce” or “acquire genes for”

Response: Line 76: Corrected as “acquire genes for”

Line 83: Change “genes of host of their innate” to “genes of the host innate”

Response: Line 83: Corrected as “genes of the host innate”

Line 84: “Dcm methylated cytosine (accession number: A0A855M2N2) possess DNA-methyltransferase activity.” One cytosine molecule cannot possess DNA-methyltransferase activity. The authors need to mention which gene has this Dcm methylated cytosine.

Response: Line 82: “genes of host innate and adaptive immune systems”

Line 86: Delete comma (,) before overexpression 

Response: Line 86: Comma has been deleted.

Line 87: Change “resistance of” to “resistance to” or “resistance against”

Response: Line 87: Corrected as “resistance to”

Line 86-88: This sentence is not clear. Suggestion: Change “have significant part of epigenetic performers” to “functions as an epigenetic performer”

Response: Line 87-88: Corrected as “functions as an epigenetic performer”

Line 91: Change “alleviates”. If this refers only to JAK2, then this is correct. If it refers to both MAPKs and JAK2, then it should be “alleviate”

Response: Line 91: Corrected as “alleviate”

Line 93: Change “In this study, it is depicted that clearly that the focus of this article aims” to “This article aims”

Response: Line 92: Corrected as “This article aims to”

Line 95: Change “play role” to “play a role”

Response: Line 93: Corrected as “play a role”

Line 95: Change “Purely, the vaccine” to “The vaccine”

Response: Line 93: Corrected as “The vaccine”

Line 96-98: “By doing computational analysis on the chosen protein sequences, the cytotoxic T-lymphocytes (CTL), B-cells, also HT-lymphocytes (HTL) epitopes are obtained.” This sentence is not clear. Suggestion: Change to “The cytotoxic T-lymphocyte (CTL), B-cell, and also HT-lymphocyte (HTL) epitopes were obtained by doing computational analysis on the chosen protein sequences.”

Response: Line 94-96: Sentence corrected as “The cytotoxic T-lymphocyte (CTL), B-cell, and also HT-lymphocyte (HTL) epitopes were obtained by doing computational analysis on the chosen protein sequences.”

Line 101-102: “Furthermore, … immunogenicity.” It is not clear what the verb is in this sentence. Is “agonists” used as a verb?

Response: Line 93-105: The respective paragraph has been revised and corrected.

Line 102-104: “The docking….worldwide” This sentence does not resemble English language.

Response: Line 100-101: The sentence has been revised.

Line 105: Change “evaluating” to “evaluated”

Response: Line 103: Corrected as suggested.

I am stopping correcting English language from this point. There are errors in every line. This is affecting the quality of the paper because it is difficult to understand what the authors are trying to say. Numerous sentences don’t have any verb.

Response: Dear reviewer, we really appreciate you line by line comments on our manuscript. We have revised the manuscript thoroughly for English proofreading.

Line 280: If the same footnote (*, ** or (***) applies to all cells in a column it can be applied to the column heading only.

Response: Line 278: Corrected as suggested.

Line 348: Table 5: While all other numbers in the table do not have any unit, for solubility, the unit needs to be specified.

Response: Table 5 (Line 347): The unit for solubility has been written in the revised manuscript.

Line 372-373: Authors please comment on the significance of these secondary structures and their implications regarding usefulness as vaccines. 

Response: Line 372-373: A sentence has been added in the revised manuscript.

Line 435: “many viral infections” Why is coronavirus, the causative agent of the current pandemic, not in the list? Also, is Ref 10 the right reference for this? It is not clear from the title of the reference.

Response: Line 436: The coronavirus has been added in the list. Furthermore, the reference has been replaced with more sittable and accurate reference.

Round 2

Reviewer 2 Report

The authors have addressed all my previous concerns and have significantly improved the language and presentation. The manuscript is of much better quality now.